# Large-scale Optimization of Partial AUC in a Range of False Positive Rates

**Yao Yao**
Department of Mathematics
The University of Iowa
`yao-yao-2@uiowa.edu`

**Qihang Lin**
Tipple College of Business
The University of Iowa
`qihang-lin@uiowa.edu`

**Tianbao Yang**
Department of Computer Science & Engineering
Texas A&M University
`tianbao-yang@tamu.edu`

## Abstract

The area under the ROC curve (AUC) is one of the most widely used performance measures for classification models in machine learning. However, it summarizes the true positive rates (TPRs) over all false positive rates (FPRs) in the ROC space, which may include the FPRs with no practical relevance in some applications. The partial AUC, as a generalization of the AUC, summarizes only the TPRs over a specific range of the FPRs and is thus a more suitable performance measure in many real-world situations. Although partial AUC optimization in a range of FPRs had been studied, existing algorithms are not scalable to big data and not applicable to deep learning. To address this challenge, we cast the problem into a non-smooth difference-of-convex (DC) program for any smooth predictive functions (e.g., deep neural networks), which allowed us to develop an efficient approximated gradient descent method based on the Moreau envelope smoothing technique, inspired by recent advances in non-smooth DC optimization. To increase the efficiency of large data processing, we used an efficient stochastic block coordinate update in our algorithm. Our proposed algorithm can also be used to minimize the sum of ranked range loss, which also lacks efficient solvers. We established a complexity of $\tilde{O}(1/\epsilon^6)$ for finding a nearly $\epsilon$-critical solution. Finally, we numerically demonstrated the effectiveness of our proposed algorithms in training both linear models and deep neural networks for partial AUC maximization and sum of ranked range loss minimization.

## 1 Introduction

The area under the receiver operating characteristic (ROC) curve (AUC) is one of the most widely used performance measures for classifiers in machine learning, especially when the data is imbalanced between the classes [7, 19]. Typically, the classifier produces a score for each data point. Then a data point is classified as positive if its score is above a chosen threshold; otherwise, it is classified as negative. Varying the threshold will change the true positive rate (TPR) and the false positive rate (FPR) of the classifier. The ROC curve shows the TPR as a function of the FPR that corresponds to the same threshold. Hence, maximizing the AUC of a classifier is essentially maximizing the classifier's average TPR over all FPRs from zero to one. However, for some applications, some FPR regions have no practical relevance. So does the TPR over those regions. For example, in clinical practice, a high FPR in diagnostic tests often results in a high monetary cost, so people may only

36th Conference on Neural Information Processing Systems (NeurIPS 2022).

need to maximize the TPR when the FPR is low [13, 34, 64]. Moreover, since two models with the same AUC can still have different ROCs, the AUC does not always reflect the true performance of a model that is needed in a particular production environment [8].

As a generalization of the AUC, the partial AUC (pAUC) only measures the area under the ROC curve that is restricted between two FPRs. A probabilistic interpretation of the pAUC can be found in [13]. In contrast to the AUC, the pAUC represents the average TPR only over a relevant range of FPRs and provides a performance measure that is more aligned with the practical needs in some applications.

In literature, the existing algorithms for training a classifier by maximizing the pAUC include the boosting method [29] and the cutting plane algorithm [41, 42, 43]. However, the former has no theoretical guarantee, and the latter applies only to linear models. More importantly, both methods require processing all the data in each iteration and thus, become computationally inefficient for large datasets.

In this paper, we proposed an approximate gradient method for maximizing the pAUC that works for nonlinear models (e.g., deep neural networks) and only needs to process randomly sampled positive and negative data points of any size in each iteration. In particular, we formulated the maximization of the pAUC as a non-smooth difference-of-convex (DC) program [30, 54]. Due to non-smoothness, most existing DC optimization algorithms cannot be applied to our formulation. Motivated by [52], we approximate the two non-smooth convex components in the DC program by their Moreau envelopes and obtain a smooth approximation of the problem, which will be solved using the gradient descent method. Since the gradient of the smooth problem cannot be calculated explicitly, we approximated the gradient by solving the two proximal-point subproblems defined by each convex component using the stochastic block coordinate descent (SBCD) method. Our method, besides its low per-iteration cost, has a rigorous theoretical guarantee, unlike the existing methods. *In fact, we show that our method finds a nearly $\epsilon$-critical point of the pAUC optimization problem in $\tilde{O}(\epsilon^{-6})$ iterations with only small samples of positive and negative data points processed per iteration.*[1] *This is the main contribution of this paper.*

Note that, for non-convex non-smooth optimization, the existing stochastic methods [10, 11] find an nearly $\epsilon$-critical point in $O(\epsilon^{-4})$ iterations under a weak convexity assumption. Our method needs $O(\epsilon^{-6})$ iterations because our problem is a DC problem with both convex components non-smooth which is much more challenging than a weakly non-convex minimization problem. In addition, our iteration number matches the known best iteration complexity for non-smooth non-convex min-max optimization [33, 46] and non-smooth non-convex constrained optimization [35].

In addition to pAUC optimization, our method can be also used to minimize the sum of ranked range (SoRR) loss, which can be viewed as a special case of pAUC optimization. Many machine learning models are trained by minimizing an objective function, which is defined as the sum of losses over all training samples [60]. Since the sum of losses weights all samples equally, it is insensitive to samples from minority groups. Hence, the sum of top-$k$ losses [17, 49] is often used as an alternative objective function because it provides the model with robustness to non-typical instances. However, the sum of top-$k$ losses can be very sensitive to outliers, especially when $k$ is small. To address this issue, [24] proposed the SoRR loss as a new learning objective, which is defined as the sum of a consecutive sequence of losses from any range after the losses are sorted. Compared to the sum of all losses and the sum of top-$k$ losses, the SoRR loss maintains a model's robustness to a minority group but also reduces the model's sensitivity to outliers. See Fig.1 in [24] for an illustration of the benefit of using the SoRR loss over other ways of aggregating individual losses.

To minimize the SoRR loss, [24] applied a difference-of-convex algorithm (DCA) [3, 54], which linearizes the second convex component and solves the resulting subproblem using the stochastic subgradient method. DCA has been well studied in literature; but when the both components are non-smooth, as in our problem, only asymptotic convergence results are available. To establish the total number of iterations needed to find an $\epsilon$-critical point in a non-asymptotic sense, most existing studies had to assume that at least one of the components is differentiable, which is not the case in this paper. Using the approximate gradient method presented in this paper, one can find a nearly $\epsilon$-critical point of the SoRR loss optimization problem in $\tilde{O}(\epsilon^{-6})$ iterations.

---

[1]Throughout the paper, $\tilde{O}(\cdot)$ suppresses all logarithmic factors.

## 2 Related Works

The pAUC has been studied for decades [26, 37, 57, 65]. However, most studies focused on its estimation [13] and application as a performance measure, while only a few studies were devoted to numerical algorithms for optimizing the pAUC. Efficient optimization methods have been developed for maximizing AUC and multiclass AUC by [69] and [66], but they cannot be applied to pAUC. Besides the boosting method [29] and the cutting plane algorithm [41, 42, 43] mentioned in the previous section, [59, 67, 68, 73] developed surrogate optimization techniques that directly maximize a smooth approximation of the pAUC or the two-way pAUC [64]. However, their approaches can only be applied when the FPR starts from exactly zero. On the contrary, our method allows the FPR to start from any value between zero and one. [61] and [47] developed algorithms that use the pAUC as a criterion for creating a linear combination of multiple existing classifiers while we consider directly train a classifier using the pAUC.

DC optimization has been studied since the 1950s [2, 20]. We refer interested readers to [30, 45, 54, 55, 58], and the references therein. The actively studied numerical methods for solving a DC program include DCA [3, 50, 54, 55], which is also known as the concave-convex procedure [32, 51, 72], the proximal DCA [5, 38, 40, 53], and the direct gradient methods [28]. However, when the two convex components are both non-smooth, the existing methods have only asymptotic convergence results except the method by [1], who considered a stopping criterion different from ours. When at least one component is smooth, non-asymptotic convergence rates have been established with and without the Kurdyka-Łojasiewicz (KL) condition [5, 6, 28, 50, 62].

The algorithms mentioned above are deterministic and require processing the entire dataset per iteration. Stochastic algorithms that process only a small data sample per iteration have been studied [12, 21, 31, 36, 44]. However, they all assumed smoothness on at least one of the two convex components in the DC program. The stochastic methods of [4, 56, 63] can be applied when both components are non-smooth but their methods require an unbiased stochastic estimation of the gradient and/or value of the two components, which is not available in the DC formulation of the pAUC maximization problem in this paper.

The technique most related to our work is the smoothing method based on the Moreau envelope [16, 18, 22, 23, 39, 52]. Our work is motivated by [39, 52], but the important difference is that they studied deterministic methods and assumed either that one function is smooth or that the proximal-point subproblems can be solved exactly, which we do not assume. However, [39, 52] consider a more general problem and study the fundamental properties of the smoothed function such as its Lipschitz smoothness and how its stationary points correspond to those of the original problems. We mainly focus on partial AUC optimization which has a special structure we can utilize when solving the proximal-point subproblems. Additionally, [52] developed an algorithm when there were linear equality constraints, which we do not consider in this paper.

## 3 Preliminary

We consider a classical binary classification problem, where the goal is to build a predictive model that predicts a binary label $y \in \{1, -1\}$ based on a feature vector $\mathbf{x} \in \mathbb{R}^p$. Let $h_{\mathbf{w}} : \mathbb{R}^p \to \mathbb{R}$ be the predictive model parameterized by a vector $\mathbf{w} \in \mathbb{R}^d$, which produces a score $h_{\mathbf{w}}(\mathbf{x})$ for $\mathbf{x}$. Then $\mathbf{x}$ is classified as positive ($y = 1$) if $h_{\mathbf{w}}(\mathbf{x})$ is above a chosen threshold and classified as negative ($y = -1$), otherwise.

Let $\mathcal{X}_+ = \{\mathbf{x}_i^+\}_i^{N_+}$ and $\mathcal{X}_- = \{\mathbf{x}_i^-\}_i^{N_-}$ be the sets of feature vectors of positive and negative training data, respectively. The problem of learning $h_{\mathbf{w}}$ through maximizing its empirical AUC on the training data can be formulated as

$$\max_{\mathbf{w}} \frac{1}{N_+ N_-} \sum_{i=1}^{N_+} \sum_{j=1}^{N_-} \mathbf{1}(h_{\mathbf{w}}(\mathbf{x}_i^+) > h_{\mathbf{w}}(\mathbf{x}_j^-)), \tag{1}$$

where $\mathbf{1}(\cdot)$ is the indicator function which equals one if the inequality inside the parentheses holds and equals zero, otherwise. According to the introduction, pAUC can be a better performance measure of $h_{\mathbf{w}}$ than AUC. Consider two FPRs $\alpha$ and $\beta$ with $0 \le \alpha < \beta \le 1$. For simplicity of exposition, we assume $N_- \alpha$ and $N_- \beta$ are both integers. Let $m = N_- \alpha$ and $n = N_- \beta$. The problem of maximizing

the empirical pAUC with FPR between $\alpha$ and $\beta$ can be formulated as

$$\max_{\mathbf{w}} \frac{1}{N_+(n-m)} \sum_{i=1}^{N_+} \sum_{j=m+1}^{n} \mathbf{1}(h_{\mathbf{w}}(\mathbf{x}_i^+) > h_{\mathbf{w}}(\mathbf{x}_{[j]}^-)), \tag{2}$$

where $[j]$ denotes the index of the $j$th largest coordinate in vector $(h_{\mathbf{w}}(\mathbf{x}_j^-))_{j=1}^{N_-}$ with ties broken arbitrarily. Note that $N_+(n-m)$ in (2) is a normalizer that makes the objective value between zero and one. Solving (2) is challenging due to discontinuity. Let $\ell : \mathbb{R} \to \mathbb{R}$ be a differential non-increasing loss function. Problem (2) can be approximated by the loss minimization problem

$$\min_{\mathbf{w}} \frac{1}{N_+(n-m)} \sum_{i=1}^{N_+} \sum_{j=m+1}^{n} \ell(h_{\mathbf{w}}(\mathbf{x}_i^+) - h_{\mathbf{w}}(\mathbf{x}_{[j]}^-)). \tag{3}$$

To facilitate the discussion, we first introduce a few notations. Given a vector $S = (s_i)_{i=1}^{N} \in \mathbb{R}^N$ and an integer $l$ with $0 \le l \le N$, the sum of the top-$l$ values in $S$ is

$$\phi_l(S) := \sum_{j=1}^{l} s_{[j]}, \tag{4}$$

where $[j]$ denotes the index of the $j$th largest coordinate in $S$ with ties broken arbitrarily. For integers $l_1$ and $l_2$ with $0 \le l_1 < l_2 \le N$, $\phi_{l_2}(S) - \phi_{l_1}(S)$ is the sum from the $(l_1+1)$th to the $l_2$th (inclusive) largest coordinates of $S$, also called a *sum of ranked range* (SoRR). In addition, we define vectors

$$S_i(\mathbf{w}) := (s_{ij}(\mathbf{w}))_{j=1}^{N_-}$$

for $i = 1, \ldots, N_+$, where $s_{ij}(\mathbf{w}) := \ell(h_{\mathbf{w}}(\mathbf{x}_i^+) - h_{\mathbf{w}}(\mathbf{x}_j^-))$ for $i = 1, \ldots, N_+$ and $j = 1, \ldots, N_-$. Since $\ell$ is non-increasing, the $j$th largest coordinate of $S_i(\mathbf{w})$ is $\ell(h_{\mathbf{w}}(\mathbf{x}_i^+) - h_{\mathbf{w}}(\mathbf{x}_{[j]}^-))$. As a result, we have, for $i = 1, \ldots, N_+$,

$$\sum_{j=m+1}^{n} \ell(h_{\mathbf{w}}(\mathbf{x}_i^+) - h_{\mathbf{w}}(\mathbf{x}_{[j]}^-)) = \phi_n(S_i(\mathbf{w})) - \phi_m(S_i(\mathbf{w})).$$

Hence, after dropping the normalizer, (3) can be equivalently written as

$$F^* = \min_{\mathbf{w}} \left\{ F(\mathbf{w}) := f^n(\mathbf{w}) - f^m(\mathbf{w}) \right\}, \tag{5}$$

where

$$f^l(\mathbf{w}) = \sum_{i=1}^{N_+} \phi_l(S_i(\mathbf{w})) \quad \text{for } l = m, n. \tag{6}$$

Next, we introduce an interesting special case of (5), namely, the problem of minimizing SoRR loss. We still consider a supervised learning problem but the target $y \in \mathbb{R}$ does not need to be binary. We want to predict $y$ based on a feature vector $\mathbf{x} \in \mathbb{R}^p$ using $h_{\mathbf{w}}(\mathbf{x})$. With a little abuse of notation, we measure the discrepancy between $h_{\mathbf{w}}(\mathbf{x})$ and $y$ by $\ell(h_{\mathbf{w}}(\mathbf{x}), y)$, where $\ell : \mathbb{R}^2 \to \mathbb{R}_+$ is a loss function. We consider learning the model's parameter $\mathbf{w}$ from a training set $\mathcal{D} = \{(\mathbf{x}_j, y_j)\}_{j=1}^{N}$, where $\mathbf{x}_j \in \mathbb{R}^p$ and $y_j \in \mathbb{R}$ for $j = 1, \ldots, N$, by minimizing the SoRR loss. More specifically, we define vector

$$S(\mathbf{w}) = (s_j(\mathbf{w}))_{j=1}^{N},$$

where $s_j(\mathbf{w}) := \ell(h_{\mathbf{w}}(\mathbf{x}_j), y_j)$, $j = 1, \ldots, N$. Recall (4). For any integers $m$ and $n$ with $0 \le m < n \le N$, the problem of minimizing the SoRR loss with a range from $m+1$ to $n$ is formulated as $\min_{\mathbf{w}} \{\phi_n(S(\mathbf{w})) - \phi_m(S(\mathbf{w}))\}$, which is an instance of (5) with

$$f^l = \phi_l(S(\mathbf{w})) \text{ for } l = m, n. \tag{7}$$

If we view $S_i(\mathbf{w})$ and $S(\mathbf{w})$ only as vector-value functions of $\mathbf{w}$ but ignore how they are formulated using data, (7) is a special case of (6) with $N_+ = 1$ and $N_- = N$.

## 4 Nearly Critical Point and Moreau Envelope Smoothing

We first develop a stochastic algorithm for (5) with $f^l$ defined in (6). To do so, we make the following assumptions, which are satisfied by many smooth $h_{\mathbf{w}}$'s and $\ell$'s.

**Assumption 1** *(a) $s_{ij}(\mathbf{w})$ is smooth and there exists $L \geq 0$ such that[2] $\|\nabla s_{ij}(\mathbf{w}) - \nabla s_{ij}(\mathbf{v})\| \leq L\|\mathbf{w} - \mathbf{v}\|$ for any $\mathbf{w}, \mathbf{v} \in \mathbb{R}^d$, $i = 1, \ldots, N_+$ and $j = 1, \ldots, N_-$. (b) There exists $B \geq 0$ such that $\|\nabla s_{ij}(\mathbf{w})\| \leq B$ for any $\mathbf{w} \in \mathbb{R}^d$, $i = 1, \ldots, N_+$ and $j = 1, \ldots, N_-$. (c) $F^* > -\infty$.*

Given $f : \mathbb{R}^d \to \mathbb{R} \cup \{+\infty\}$, the subdifferential of $f$ is

$$\partial f(\mathbf{w}) = \left\{ \boldsymbol{\xi} \in \mathbb{R}^d \left| f(\mathbf{v}) \geq h(\mathbf{w}) + \boldsymbol{\xi}^\top (\mathbf{v} - \mathbf{w}) + o(\|\mathbf{v} - \mathbf{w}\|_2), \ \mathbf{v} \to \mathbf{w} \right. \right\},$$

where each element in $\partial f(\mathbf{w})$ is called a subgradient of $f$ at $\mathbf{w}$. We say $f$ is $\rho$-**weakly convex** for some $\rho \geq 0$ if $f(\mathbf{v}) \geq f(\mathbf{w}) + \langle \boldsymbol{\xi}, \mathbf{v} - \mathbf{w} \rangle - \frac{\rho}{2}\|\mathbf{v} - \mathbf{w}\|^2$ for any $\mathbf{v}$ and $\mathbf{w}$ and $\boldsymbol{\xi} \in \partial f(\mathbf{w})$ and say $f$ is $\rho$-**strongly convex** for some $\rho \geq 0$ if $f(\mathbf{v}) \geq f(\mathbf{w}) + \langle \boldsymbol{\xi}, \mathbf{v} - \mathbf{w} \rangle + \frac{\rho}{2}\|\mathbf{v} - \mathbf{w}\|^2$ for any $\mathbf{v}$ and $\mathbf{w}$ and $\boldsymbol{\xi} \in \partial f(\mathbf{w})$. It is known that, if $f$ is $\rho$-weakly convex, then $f(\mathbf{w}) + \frac{1}{2\mu}\|\mathbf{w}\|^2$ is a $(\mu^{-1} - \rho)$-strongly convex function when $\mu^{-1} > \rho$.

Under Assumption 1, $\phi_l(S_i(\mathbf{w}))$ is a composite of the closed convex function $\phi_l$ and the smooth map $S_i(\mathbf{w})$. According to Lemma 4.2 in [14], we have the following lemma.

**Lemma 1** *Under Assumption 1, $f^m(\mathbf{w})$ and $f^n(\mathbf{w})$ in (6) are $\rho$-weakly convex with $\rho := N_+ N_- L$.*

To solve (5) numerically, we need to overcome the following challenges. (i) $F(\mathbf{w})$ is non-convex even if each $s_{ij}(\mathbf{w})$ is convex. In fact, $F(\mathbf{w})$ is a DC function because, by Lemma 1, we can represent $F(\mathbf{w})$ as the difference of the convex functions $f^n(\mathbf{w}) + \frac{1}{2\mu}\|\mathbf{w}\|^2$ and $f^m(\mathbf{w}) + \frac{1}{2\mu}\|\mathbf{w}\|^2$ with $\mu^{-1} > \rho$. (ii) $F(\mathbf{w})$ is non-smooth due to $\phi_l$ so that finding an approximate critical point (defined below) of $F(\mathbf{w})$ is difficult. (iii) Computing the exact subgradient of $f^l(\mathbf{w})$ for $l = m, n$ requires processing $N_+ N_-$ data pairs, which is computationally expensive for a large data set.

Because of challenges (i) and (ii), we have to consider a reasonable goal when solving (5). We say $\mathbf{w} \in \mathbb{R}^d$ is a *critical point* of (5) if $\mathbf{0} \in \partial f^n(\mathbf{w}) - \partial f^m(\mathbf{w})$. Given $\epsilon > 0$, we say $\mathbf{w} \in \mathbb{R}^d$ is an $\epsilon$-*critical point* of (5) if there exists $\boldsymbol{\xi} \in \partial f^n(\mathbf{w}) - \partial f^m(\mathbf{w})$ such that $\|\boldsymbol{\xi}\| \leq \epsilon$. A critical point can only be achieved asymptotically in general.[3] Within finitely many iterations, there also exists no algorithm that can find an $\epsilon$-critical point unless at least one of $f^m$ and $f^n$ is smooth, e.g., [63]. Since $f^m$ and $f^n$ are both non-smooth, we have to consider a weaker but achievable target, which is a nearly $\epsilon$-critical point defined below.

**Definition 1** *Given $\epsilon > 0$, we say $\mathbf{w} \in \mathbb{R}^d$ is a nearly $\epsilon$-critical point of (5) if there exist $\boldsymbol{\xi}$, $\mathbf{w}'$, and $\mathbf{w}'' \in \mathbb{R}^d$ such that $\boldsymbol{\xi} \in \partial f^n(\mathbf{w}') - \partial f^m(\mathbf{w}'')$ and $\max\{\|\boldsymbol{\xi}\|, \|\mathbf{w} - \mathbf{w}'\|, \|\mathbf{w} - \mathbf{w}''\|\} \leq \epsilon$.*

Definition 1 is reduced to the $\epsilon$-stationary point defined by [39, 52] when $\mathbf{w}$ equals $\mathbf{w}'$ or $\mathbf{w}''$. However, obtaining their $\epsilon$-stationary point requires exactly solving the proximal mapping of $f^m$ or $f^n$ while finding a nearly $\epsilon$-critical point requires only solving the proximal mapping inexactly. When $\mathbf{w}$ is generated by a stochastic algorithm, we also call $\mathbf{w}$ a nearly $\epsilon$-critical point if it satisfies Definition 1 with each $\|\cdot\|$ replaced by $\mathbb{E}\|\cdot\|$.

Motivated by [52] and [39], we approximate non-smooth $F(\mathbf{w})$ by a smooth function using the Moreau envelopes. Given a proper, $\rho$-weakly convex and closed function $f$ on $\mathbb{R}^d$, the *Moreau envelope* of $f$ with the smoothing parameter $\mu \in (0, \rho^{-1})$ is defined as

$$f_\mu(\mathbf{w}) := \min_{\mathbf{v}} \left\{ f(\mathbf{v}) + \frac{1}{2\mu}\|\mathbf{v} - \mathbf{w}\|^2 \right\} \tag{8}$$

and the *proximal mapping* of $f$ is defined as

$$\mathbf{v}_{\mu f}(\mathbf{w}) := \arg\min_{\mathbf{v}} \left\{ f(\mathbf{v}) + \frac{1}{2\mu}\|\mathbf{v} - \mathbf{w}\|^2 \right\}. \tag{9}$$

Note that the $\mathbf{v}_{\mu f}(\mathbf{w})$ is unique because the minimization above is strongly convex. Standard results show that $f_\mu(\mathbf{w})$ is smooth with $\nabla f_\mu(\mathbf{w}) = \mu^{-1}(\mathbf{w} - \mathbf{v}_{\mu f}(\mathbf{w}))$ and $\mathbf{v}_{\mu f}(\mathbf{w})$ is $(1 - \mu\rho)^{-1}$-Lipschitz continuous. See Proposition 13.37 in [48] and Proposition 1 in [52]. Hence, using the Moreau envelope, we can construct a smooth approximation of (5) as follows

$$\min_{\mathbf{w}} \left\{ F^\mu := f_\mu^n(\mathbf{w}) - f_\mu^m(\mathbf{w}) \right\}. \tag{10}$$

---

[2]In this paper, $\|\cdot\|$ represents Euclidean norm.

[3]A stronger notion than criticality is (directional) stationarity, which can also be achieved asymptotically [45].

Function $F^\mu$ has the following properties. The first property is shown in [52]. We give the proof for the second in Appendix B.

**Lemma 2** *Suppose Assumption 1 holds and $\mu > \rho^{-1}$ with $\rho$ defined in Lemma 1. The following claims hold*

1. $\nabla F^\mu(\mathbf{w}) = \mu^{-1}(\mathbf{v}_{\mu f^m}(\mathbf{w}) - \mathbf{v}_{\mu f^n}(\mathbf{w}))$ *and it is $L_\mu$-Lipschitz continuous with $L_\mu = \frac{2}{\mu - \mu^2 \rho}$.*

2. *If $\bar{\mathbf{v}}$ and $\mathbf{w}$ are two random vectors such that $\mathbb{E}\|\nabla F^\mu(\mathbf{w})\|^2 \leq \min\{1, \mu^{-2}\}\epsilon^2/4$ and $\mathbb{E}\|\bar{\mathbf{v}} - \mathbf{v}_{\mu f^l}(\mathbf{w})\|^2 \leq \epsilon^2/4$ for either $l = m$ or $l = n$, then $\bar{\mathbf{v}}$ is a nearly $\epsilon$-critical points of (5).*

Since $F^\mu$ is smooth, we can directly apply a first-order method for smooth non-convex optimization to (10). To do so, we need to evaluate $\nabla F^\mu(\mathbf{w})$, which requires computing $\mathbf{v}_{\mu f^m}(\mathbf{w})$ and $\mathbf{v}_{\mu f^n}(\mathbf{w})$, i.e., exactly solving (9) with $f = f^m$ and $f = f^n$, respectively. Computing the subgradients of $f^m$ and $f^n$ require processing $N_+ N_-$ data pairs which is costly. Unfortunately, the standard approach of sampling over data pairs does not produce unbiased stochastic subgradients of $f^m$ and $f^n$ due to the composite structure $\phi_l(S_i(\mathbf{w}))$. In the next section, we will discuss a solution to overcome this challenge and approximate $\mathbf{v}_{\mu f^m}(\mathbf{w})$ and $\mathbf{v}_{\mu f^n}(\mathbf{w})$, which leads to an efficient algorithm for (10).

# 5 Algorithm for pAUC Optimization

Consider (10) with $f^l$ defined in (6) for $l = m$ and $n$. To avoid of processing $N_+ N_-$ data points, one method is to introduce dual variables $\mathbf{p}_i = (p_{ij})_{j=1}^{N_-}$ for $i = 1, \ldots, N_+$ and formulate $f^l$ as

$$f^l(\mathbf{w}) = \max_{\mathbf{p}_i \in \mathcal{P}^l, i=1,\ldots,N_+} \left\{ \sum_{i=1}^{N_+} \sum_{j=1}^{N_-} p_{ij} s_{ij}(\mathbf{w}) \right\}, \tag{11}$$

where $\mathcal{P}^l = \{\mathbf{p} \in \mathbb{R}^{N_-} | \sum_{j=1}^{N_-} p_j = l, \ p_j \in [0,1]\}$. Then (10) can be reformulated as a min-max problem and solved by a primal-dual stochastic gradient method (e.g. [46]). However, the maximization in (11) involves $N_+ N_-$ decision variables and equality constraints, so the per-iteration cost is still $O(N_+ N_-)$ even after using stochastic gradients.

To further reduce the per-iteration cost, we take the dual form of the maximization in (11) (see Lemma 4 in Appendix B) and formulate $f^l$ as

$$f^l(\mathbf{w}) = \min_{\boldsymbol{\lambda}} \left\{ g^l(\mathbf{w}, \boldsymbol{\lambda}) := l\mathbf{1}^\top \boldsymbol{\lambda} + \sum_{i=1}^{N_+} \sum_{j=1}^{N_-} [s_{ij}(\mathbf{w}) - \lambda_i]_+ \right\}, \tag{12}$$

where $\boldsymbol{\lambda} = (\lambda_1, \ldots, \lambda_{N_+})$. Hence, (9) with $f = f^l$ for $l = m$ and $n$ can be reformulated as

$$\min_{\mathbf{v}, \boldsymbol{\lambda}} \left\{ g^l(\mathbf{v}, \boldsymbol{\lambda}) + \frac{1}{2\mu}\|\mathbf{v} - \mathbf{w}\|^2 \right\}. \tag{13}$$

Note that $g^l(\mathbf{v}, \boldsymbol{\lambda})$ is jointly convex in $\mathbf{v}$ and $\boldsymbol{\lambda}$ when $\mu^{-1} > \rho = N_+ N_- L$ (see Lemma 3 in Appendix B). Thanks to formulation (13), we can construct stochastic subgradient of $g^l$ and apply coordinate update to $\boldsymbol{\lambda}$ by sampling indexes $i$'s and $j$'s, which significantly reduce the computational cost when $N_+$ and $N_-$ are both large. We present this standard stochastic block coordinate descent (SBCD) method for solving (13) in Algorithm 1 and present its convergence property as follows.

**Proposition 1** *Suppose Assumption 1 holds and $\mu^{-1} > \rho = N_+ N_- L$, $\theta_t = \frac{\text{dist}(\boldsymbol{\lambda}^{(0)}, \Lambda^*)}{\sqrt{IT} N_-}$ and $\eta_t = \frac{\|\mathbf{v}_{\mu f^l}(\mathbf{w}) - \mathbf{w}\|}{N_+ N_- B \sqrt{T}}$ for any $t$ in Algorithm 1. It holds that*

$$\left( \frac{1}{2\mu} - \frac{\rho}{2} \right) \mathbb{E}\|\bar{\mathbf{v}} - \mathbf{v}_{\mu f^l}(\mathbf{w}))\|^2 \leq \frac{N_+ N_-}{\sqrt{IT}} \text{dist}(\boldsymbol{\lambda}^{(0)}, \Lambda^*) + \frac{N_+ N_- B}{2\sqrt{T}} \|\mathbf{v}_{\mu f^l}(\mathbf{w}) - \mathbf{w}\| + \frac{\|\mathbf{v}_{\mu f^l}(\mathbf{w}) - \mathbf{w}\|^2}{2\mu T},$$

*where $\Lambda^* = \arg\min_{\boldsymbol{\lambda}} g^l(\mathbf{v}_{\mu f^l}(\mathbf{w}), \boldsymbol{\lambda})$.*

Using Algorithm 1 to compute an approximation of $\mathbf{v}_{\mu f^l}(\mathbf{w})$ for $l = m$ and $n$ and thus, an approximation of $\nabla F^\mu(\mathbf{w})$, we can apply an approximate gradient descent (AGD) method to (10) and find a nearly $\epsilon$-critical point of (5) according to Lemma 2. We present the AGD method in Algorithm 2 and its convergence property as follows.

**Algorithm 1** Stochastic Block Coordinate Descent for (13): $(\bar{\mathbf{v}}, \bar{\boldsymbol{\lambda}}) = \text{SBCD}(\mathbf{w}, \boldsymbol{\lambda}, T, \mu, l)$

1: **Input:** Initial solution $(\mathbf{w}, \boldsymbol{\lambda})$, the number of iterations $T$, $\mu > 0$, an integer $l > 0$ and sample sizes $I$ and $J$.
2: Set $(\mathbf{v}^{(0)}, \boldsymbol{\lambda}^{(0)}) = (\mathbf{w}, \boldsymbol{\lambda})$ and choose $(\eta_t, \theta_t)_{t=0}^{T-1}$.
3: **for** $t = 0$ **to** $T - 1$ **do**
4:     Sample $\mathcal{I}_t \subset \{1, \ldots, N_+\}$ with $|\mathcal{I}_t| = I$ and sample $\mathcal{J}_t \subset \{1, \ldots, N_-\}$ with $|\mathcal{J}_t| = J$.
5:     Compute stochastic subgradient w.r.t. $\mathbf{v}$:
$$G_{\mathbf{v}}^{(t)} = \frac{N_+ N_-}{IJ} \sum_{i \in \mathcal{I}_t} \sum_{j \in \mathcal{J}_t} \nabla s_{ij}(\mathbf{v}^{(t)}) \mathbf{1}\left(s_{ij}(\mathbf{v}^{(t)}) > \lambda_i^{(t)}\right)$$
6:     Proximal stochastic subgradient update on $\mathbf{v}$:
$$\mathbf{v}^{(t+1)} = \arg\min_{\mathbf{v}} (G_{\mathbf{v}}^{(t)})^\top \mathbf{v} + \frac{\|\mathbf{v} - \mathbf{w}\|^2}{2\mu} + \frac{\|\mathbf{v} - \mathbf{v}^{(t)}\|^2}{2\eta_t} \qquad (14)$$
7:     Compute stochastic subgradient w.r.t. $\lambda_i$ for $i \in \mathcal{I}_t$:
$$G_{\lambda_i}^{(t)} = l - \frac{N_-}{J} \sum_{j \in \mathcal{J}_t} \mathbf{1}\left(s_{ij}(\mathbf{v}^{(t)}) > \lambda_i^{(t)}\right) \text{ for } i \in \mathcal{I}_t$$
8:     Stochastic block subgradient update on $\lambda_i$ for $i \in \mathcal{I}_t$:
$$\lambda_i^{(t+1)} = \lambda_i^{(t)} - \theta_t G_{\lambda_i}^{(t)} \text{ for } i \in \mathcal{I}_t \quad \text{and} \quad \lambda_i^{(t+1)} = \lambda_i^{(t)} \text{ for } i \notin \mathcal{I}_t. \qquad (15)$$
9: **end for**
10: **Output:** $(\bar{\mathbf{v}}, \bar{\boldsymbol{\lambda}}) = \frac{1}{T} \sum_{t=0}^{T-1} (\mathbf{v}^{(t)}, \boldsymbol{\lambda}^{(t)})$.

---

**Algorithm 2** Approximate Gradient Descent (AGD) for (10)

1: **Input:** Initial solutions $(\mathbf{w}^{(0)}, \bar{\boldsymbol{\lambda}}_m^{(0)}, \bar{\boldsymbol{\lambda}}_n^{(0)})$, the number of iterations $K$, $\mu > \rho^{-1}$, $\gamma > 0$, $m = \alpha N_-$ and $n = \beta N_-$.
2: **for** $k = 0$ **to** $K - 1$ **do**
3:     $(\bar{\mathbf{v}}_m^{(k)}, \bar{\boldsymbol{\lambda}}_m^{(k+1)}) = \text{SBMD}(\mathbf{w}^{(k)}, \bar{\boldsymbol{\lambda}}_m^{(k)}, T_k, \mu, m)$
4:     $(\bar{\mathbf{v}}_n^{(k)}, \bar{\boldsymbol{\lambda}}_n^{(k+1)}) = \text{SBMD}(\mathbf{w}^{(k)}, \bar{\boldsymbol{\lambda}}_n^{(k)}, T_k, \mu, n)$
5:     $\mathbf{w}^{(k+1)} = \mathbf{w}^{(k)} - \gamma \mu^{-1}(\bar{\mathbf{v}}_m^{(k)} - \bar{\mathbf{v}}_n^{(k)})$
6: **end for**
7: **Output:** $\bar{\mathbf{v}}_n^{(\bar{k})}$ with $\bar{k}$ sampled from $\{0, \ldots, K - 1\}$.

---

**Theorem 1** *Suppose Assumption 1 holds and Algorithm 1 is called in iteration $k$ of Algorithm 2 with parameters $\mu^{-1} > \rho = N_+ N_- L$, $\theta_t = \frac{dist(\bar{\boldsymbol{\lambda}}^{(k)}, \Lambda_k^*)}{\sqrt{IT_k} N_-}$, $\eta_t = \frac{\|\mathbf{v}_{\mu f^l}(\mathbf{w}^{(k)}) - \mathbf{w}^{(k)}\|}{N_+ N_- B \sqrt{T_k}}$ for any $t$, and*
$$T_k = \max\left\{\frac{144 N_+^2 N_-^2 D_l^2 (k+1)^2}{I(\mu^{-1} - \rho)^2}, \frac{4 N_+^2 N_-^2 \mu^2 l^2 B^2 (k+1)^2}{(\mu^{-1} - \rho)^2}, \frac{6\mu l^2 B^2 (k+1)}{2(\mu^{-1} - \rho)^2}\right\}$$
*where $\Lambda_k^* = \arg\min_{\boldsymbol{\lambda}} g^l(\mathbf{v}_{\mu f^l}(\mathbf{w}^{(k)}), \boldsymbol{\lambda})$ and*
$$D_l := \max\left\{dist(\bar{\boldsymbol{\lambda}}_l^{(0)}, \Lambda_0^*), \quad \frac{1}{2}\left(\frac{1}{\mu} - \rho\right) + \frac{\mu l^2 B^2}{2} + N_+ B + \frac{N_+ B}{1 - \mu\rho}\left(\frac{2\gamma}{\mu} + \gamma n B + \gamma m B\right)\right\}. \quad (16)$$
*Then $\bar{\mathbf{v}}_n^{(\bar{k})}$ is a nearly $\epsilon$-critical point of (5) with $f^l$ defined in (6) with $K$ no more than*
$$K = \max\left\{\frac{16\mu^2}{\gamma \min\{1, \mu^2\}\epsilon^2}\left(F(\mathbf{v}_{\mu f^n}(\mathbf{w}^{(0)})) - F^*\right), \frac{96}{\min\{1, \mu^2\}\epsilon^2} \log\left(\frac{96}{\min\{1, \mu^2\}\epsilon^2}\right)\right\}. \quad (17)$$

According to Theorem 1, to find a nearly $\epsilon$-critical point of (5), we need $K = \tilde{O}(\epsilon^{-2})$ iterations in Algorithm 2 and $\sum_{k=0}^{K-1} T_k = O(K^3) = \tilde{O}(\epsilon^{-6})$ iterations of Algorithm 1 in total across all calls.

**Remark 1 (Challenges in proving Theorem 1)** *Suppose we can set $T_k$ in lines 3 and 4 of Algorithm 2 appropriately such that the approximation errors $\mathbb{E}\|\bar{\mathbf{v}}_m^{(k)} - \mathbf{v}_{\mu f^m}(\mathbf{w}^{(k)})\|^2$ and $\mathbb{E}\|\bar{\mathbf{v}}_n^{(k)} -$*

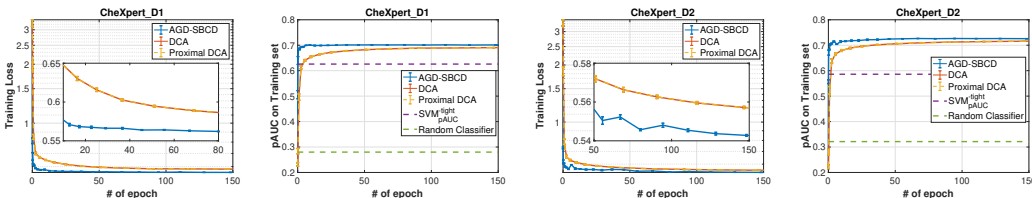

Figure 1: Results for Patial AUC Maximization of D1 and D2. (Results of D3, D4 and D5 are shown in Appendix E.3 Figure 3)

$\mathbf{v}_{\mu f^n}(\mathbf{w}^{(k)})\|^2$ are both $O(1/k)$. We can then prove that Algorithm 2 finds a nearly $\epsilon$-critical point within $K = \tilde{O}(\epsilon^{-2})$ iterations and the total complexity is $\sum_{k=0}^{K-1} T_k$. This is just a standard idea. However, by Proposition 1, such a $T_k$ must be $\Theta(k^2(dist^2(\bar{\boldsymbol{\lambda}}^{(k)}, \Lambda_k^*) + \|\mathbf{v}_{\mu f^l}(\mathbf{w}^{(k)}) - \mathbf{w}^{(k)}\|^2))$ where $dist^2(\bar{\boldsymbol{\lambda}}^{(k)}, \Lambda_k^*)$ and $\|\mathbf{v}_{\mu f^l}(\mathbf{w}^{(k)}) - \mathbf{w}^{(k)}\|^2$ also change with $k$. Then it is not clear what the order of $T_k$ is. By a novel proving technique based on the (linear) error-bound condition of $g^l(\mathbf{w}, \boldsymbol{\lambda})$ with respect to $\boldsymbol{\lambda}$, we prove that both $dist^2(\bar{\boldsymbol{\lambda}}^{(k)}, \Lambda_k^*)$ and $\|\mathbf{v}_{\mu f^l}(\mathbf{w}^{(k)}) - \mathbf{w}^{(k)}\|^2$ are $O(1)$ (see (27) and (30) in Appendix D) which ensures that $T_k = \Theta(k^2)$ and thus the total complexity is $\sum_{k=0}^{K-1} T_k = O(K^3) = \tilde{O}(\epsilon^{-6})$.

**Remark 2 (Analysis of sensitivity of the algorithm to $\mu$)** *For the interesting case where $\rho \geq 1$, we have $\mu < 1/\rho < 1$. In this case, we can derive that the order of dependency on $\mu$ is $O(\frac{1}{\epsilon^6 \mu^6})$ and the optimal choice of $\mu$ is thus $\Theta(\rho^{-1})$, e.g., $\mu = \frac{1}{2\rho}$, which leads to a complexity of $O(\rho^6/\epsilon^6)$. We present the convergence curves and the test performance of our method when applied to training a linear model with $\mu$ of different values in Appendix E.7.*

The technique in the previous sections can be directly applied to minimize the SoRR loss, which is formulated as (5) but with $f^l$ defined in (7). Due to the limit of space, we present the algorithm for minimizing the SoRR loss and its convergence result in Appendix A.

## 6 Numerical Experiments

In this section, we demonstrate the effectiveness of our algorithm AGD-SBCD for pAUC maximization and SoRR loss minimization problems (see Appendix E.1 for details). All experiments are conducted in Python and Matlab on a computer with the CPU 2GHz Quad-Core Intel Core i5 and the GPU NVIDIA GeForce RTX 2080 Ti. All datasets we used are publicly available and contain no personally identifiable information and offensive contents.

### 6.1 Partial AUC Maximization

For maximizing pAUC, we focus on large-scale imbalanced medical dataset CheXpert [25], which is licensed under CC-BY-SA and has 224,316 images. We construct five binary classification tasks with the logistic loss $\ell(z) = \log(1 + \exp(-z))$ for predicting five popular diseases, Cardiomegaly (D1), Edema (D2), Consolidation (D3), Atelectasis (D4), and P. Effusion (D5).

For comparison of training convergence, we consider different methods for optimizing the partial AUC. We compare with three baselines, DCA [24] (see Appendix E.4 for details), proximal DCA [62] (see Appendix E.5 for details) and SVM$_{pAUC}$-tight [42]. Since DCA, proximal DCA and SVM$_{pAUC}$-tight cannot be applied to deep neural networks, we focus on linear model and use a pre-trained deep neural network to extract a fixed dimensional feature vectors of 1024. The deep neural network was trained by optimizing the cross-entropy loss following the same setting as in [70].

For three baselines and our algorithm, the process to tune the hyper-parameters is explained in Appendix E.2. In Figure 1 and Figure 3 in Appendix E.3, we show how the training loss (the objective value of (3)) and normalized partial AUC on the training data change with the number of epochs.

Table 1: Comparison on the CheXpert training data. From left to right, the columns are the tasks, the pAUCs returned by SVM$_{pAUC}$-tight, the CPU time (in seconds) SVM$_{pAUC}$-tight takes, the CPU and GPU time AGD-SBCD uses to exceed SVM$_{pAUC}$-tight's pAUCs, the final pAUCs returned by AGD-SBCD, and the CPU and GPU time (in seconds) AGD-SBCD takes to return the final pAUCs.

| Methods | SVM$_{pAUC}$-tight | | AGD-SBCD | | | | |
|---|---|---|---|---|---|---|---|
| Tasks | pAUC | CPU time | CPU time (epoch) to outperform | GPU time to outperform | pAUC | CPU time | GPU time |
| D1 | 0.6259 | 95.14 | 2.91 (0.23) | 1.85 | 0.7005±0.0003 | 118.32 | 82.13 |
| D2 | 0.5860 | 90.83 | 3.36 (0.23) | 1.93 | 0.7214±0.0024 | 415.66 | 247.29 |
| D3 | 0.3745 | 90.56 | 3.26 (0.23) | 1.84 | 0.4910±0.0006 | 181.70 | 104.55 |
| D4 | 0.3895 | 89.64 | 10.09 (0.63) | 8.38 | 0.4616±0.0006 | 187.36 | 158.14 |
| D5 | 0.7267 | 90.86 | 3.97 (0.23) | 1.89 | 0.8272±0.0001 | 238.10 | 142.91 |

We observe that for all of these five diseases, our algorithm converges much faster than DCA and proximal DCA and we get a better partial AUC than DCA and proximal DCA.

The comparison between our AGD-SBCD and SVM$_{pAUC}$-tight on training data are shown in Table 1. As shown from the second to the fifth column of Table 1, our algorithm needs only a few seconds to exceed the pAUCs that SVM$_{pAUC}$-tight takes more than one minute to return. As shown from sixth to eighth column, our algorithm eventually improves the pAUC by at least 12% compared with SVM$_{pAUC}$-tight. DCA and proximal DCA are not included in the tables because it computes deterministic subgradients, which leads to a runtime significantly longer than the other two methods. We plot the convergence curves of training pAUC over GPU time for DCA and our algorithm in Figure 7 in Appendix E.8.

To compare the testing performances, we consider the deep neural networks besides the linear model. For linear model, we still compare with DCA and SVM$_{pAUC}$-tight. For deep neural networks, we compare with the naive mini-batch based method (MB) [27] and methods based on different optimization objectives, including the cross-entropy loss (CE) and the AUC min-max margin loss (AUC-M) [71]. We learn the model DenseNet121 from scratch with the CheXpert training data split in train/val=9:1 and the CheXpert validation dataset as the testing set, which has 234 samples. The range of FPRs in pAUC is [0.05, 0.5]. For optimizing CE, we use the standard Adam optimizer. For optimizing AUC-M, we use the PESG optimizer in [71]. We run each method 10 epochs and the learning rate ($c$ in AGD-SBCD) of all methods is tuned from $\{10^{-5} \sim 10^0\}$. The mini-batch size is 32. For AGD-SBCD, $T_k$ is set to $50(k+1)^2$, $\mu$ is set to $\frac{10^3}{N_+ N_-}$ and $\gamma$ is tuned from $\{0.1, 1, 2\} \times 10^3 / (N_+ N_-)$. For MB, the learning rate decays in the same way as in [27]. For CE and AUC-M, the learning rate decays 10-fold after every 5 epochs. For AUC-M, we tune the hyperparameter $\gamma$ in $\{100, 500, 1000\}$. For each method, the validation set is used to tune the hyperparameters and select the best model across all iterations. The results of the pAUCs on the testing set are reported in Table 2, which shows that our algorithm performs the best for all diseases. The complete ROC curves on the testing set are shown in Appendix E.3.

Table 2: The pAUCs with FPRs between 0.05 and 0.5 on the testing sets from the CheXpert data.

| | Method | D1 | D2 | D3 | D4 | D5 |
|---|---|---|---|---|---|---|
| Linear Model | SVM$_{pAUC}$-tight | 0.6538±0.0042 | 0.6038±0.0009 | 0.6946±0.0020 | **0.6521±0.0006** | 0.7994±0.0004 |
| | DCA | 0.6636±0.0093 | 0.8078±0.0030 | 0.7427±0.0257 | 0.6169±0.0208 | 0.8371±0.0022 |
| | Proximal DCA | 0.6615±0.0103 | 0.8041±0.0033 | 0.7064±0.0253 | 0.5945±0.0266 | 0.8352±0.0023 |
| | AGD-SBCD | **0.6721±0.0081** | **0.8257±0.0025** | **0.8016±0.0075** | 0.6340±0.0165 | **0.8500±0.0017** |
| Deep Model | MB | 0.7510±0.0248 | 0.8197±0.0127 | 0.6339±0.0328 | 0.5698±0.0343 | 0.8461±0.0188 |
| | CE | 0.6994±0.0453 | 0.8075±0.0244 | 0.7673±0.0266 | 0.6499±0.0184 | 0.7884±0.0080 |
| | AUC-M | 0.7403±0.0339 | 0.8002±0.0274 | 0.8533±0.0469 | 0.7420±0.0277 | 0.8504±0.0065 |
| | AGD-SBCD | **0.7535±0.0255** | **0.8345±0.0130** | **0.8689±0.0184** | **0.7520±0.0079** | **0.8513±0.0107** |

For deep neural networks, we also learn the model ResNet-20 from scratch with the CIFAR-10-LT and the Tiny-ImageNet-200-LT datasets, which are constructed similarly as in [67]. Details about these two datasets are summarized in Appendix E.6. The range of FPRs in pAUC is [0.05, 0.5]. The process of tuning hyperparameters is the same as that for CheXpert. The results of the pAUCs on the

testing set are reported in Table 3, which shows that our algorithm performs the best for these two long-tailed datasets.

Table 3: The pAUCs with FPRs between $0.05$ and $0.5$ on the testing sets from the CIFAR-10-LT and the Tiny-ImageNet-200-LT Datasets.

| | Dataset | MB | CE | AUC-M | AGD-SBCD |
|---|---|---|---|---|---|
| Deep Model | CIFAR-10-LT | $0.9337\pm0.0043$ | $0.9016\pm0.0137$ | $0.9323\pm0.0055$ | $\mathbf{0.9408\pm0.0084}$ |
| | Tiny-ImageNet-200-LT | $0.6445\pm0.0214$ | $0.6549\pm0.008$ | $0.6497\pm0.009$ | $\mathbf{0.6594\pm0.0192}$ |

## 7  Conclusion

Most existing methods for optimizing pAUC are deterministic and only have an asymptotic convergence property. We formulate pAUC optimization as a non-smooth DC program and develop a stochastic subgradient method based on the Moreau envelope smoothing technique. We show that our method finds a nearly $\epsilon$-critical point in $\tilde{O}(\epsilon^{-6})$ iterations and demonstrate its performance numerically. A limitation of this paper is the smoothness assumption on $s_{ij}(\mathbf{w})$, which does not hold for some models, e.g., neural networks using ReLU activation functions. It is a future work to extend our results for non-smooth models.

## Acknowledgements

This work was jointly supported by the University of Iowa Jumpstarting Tomorrow Program and NSF award 2147253. T. Yang was also supported by NSF awards 2110545 and 1844403, and Amazon research award. We thank Zhishuai Guo, Zhuoning Yuan and Qi Qi for discussing about processing the image dataset.

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
