# OpenReview forum: "Large-scale Optimization of Partial AUC in a Range of False Positive Rates"
_NeurIPS.cc/2022/Conference — NeurIPS 2022 Accept_

### Official Review · Reviewer_412K · 2022-07-07

**Rating:** 6
**Confidence:** 3
**Soundness:** 2 fair
**Presentation:** 2 fair
**Contribution:** 2 fair

**Summary:**

This paper proposed the method for partial AUC optimization. Compared with [24] and [42], the proposed method can be applied to deep neural networks. The idea of the proposed method is to formulate the maximization of pAUC as a non-smooth difference-of-convex (DC) program and to apply the Moreau envelop technique. Furthermore, the stochastic block coordinate descent (SBCD) method is used for handling large-scale data. The theoretical analysis showed the complexity of finding a nearly \epsilon-critical solution. Finally, the performance of the proposed method was compared with the existing pAUC methods.

**Questions:**

The main point of this paper seems to represent the pAUC objective function as DoC. The theoretical analysis seems based on the DoC property. The empirical evaluation showed that the performance of the proposed method is superior to the existing methods. However, my concern is how this approach is meaningful from the theoretical viewpoint. To represent the pAUC as DoC, it is necessary to add $1/(2\mu)||w||^2$ to non-convex functions $f^n(w)$ and $f^m(w)$. Along with this line, one can prove, e.g., the convergence property of their method with deep neural networks plus $1/(2\mu)||w||^2$ regularization (i.e., weight-decay) on the basis of convex analysis. Compared with such an analysis, how different this paper is? In addition, to make $f^n(w) + 1/(2\mu)||w||^2$ convex, $\mu$ needs to be too small. In this case, the constants in the complexity order finding a nearly $\epsilon$-critical point.

In Table 1, the CPU and GPU time AGD-SBCD uses to exceed SVM_pAUC^tight's pAUCs were reported. It seems good. A small concern is that optimization algorithms sometimes take most of the computation to check convergence even though the method attains some level of performance. If the authors could include the graph plots pAUC on a vertial axis and CPU+GPU time on a horizontal axis, such a graph might help understand the performance of the proposed method. One can check that performance curves do not drop after exceeding the existing method's pAUCs.

The proposed method took a few seconds to exceed the SVM_pAUC^tight's pAUCs. Probably, the proposed method outperforms less than one epoch without going through scanning all training samples. If so, are there any explanations for why this can happen? Other than computation time, is it possible to report the number of epochs or iterations to exceed the SVM_pAUC^tight's pAUCs?

**Limitations:**

The limitation about the smoothness assumption was discussed in Conclusion.

**Strengths And Weaknesses:**

##### Strengths
- The proposed method enables us to train deep neural networks
- The proposed method scales well because of stochastic optimization

##### Weaknesses
- $\tilde{O}(\epsilon^-6)$ iterations seems large
- $\mu$ needs to be small to make $f(w) + 1/(2\mu)||w||^2$ convex, resulting in that the constant in the order of K becomes very large

---

> ### Author Response · Authors · 2022-08-02
> **Author response to Reviewer 412K**
>
> Thank you for your time and comments. The following are responses to your questions and concerns. The equation and reference numbers we mentioned below follow the rebuttal revision of the paper.
>
> **1. $\tilde{O}(\epsilon^{-6})$ iterations seems large.**
>
> **A**: This is the best complexity in literature for DC programs with both components non-smooth. It is large due to the difficulty of the problem, especially the non-smoothness. We think it is unlikely that this complexity can be further reduced. In fact, the  best known complexity for non-smooth weakly convex min-max optimization is $\tilde{O}(\epsilon^{-6})$ by [46], and our problem is even more challenging than [46] since a weakly convex function minus a weakly convex function is not weakly convex.
>
> **2. $\mu$ needs to be small to make $f(w)+\frac{1}{2\mu}\\|w\\|^2$ convex, resulting in that the constant in the order of K becomes very large.**
>
> **A**: The value of $\mu$ only depends on the weak convexity parameter of $f^m$ and $f^n$ which is a problem-dependent constant (just like the Lipschitz constant) and is not necessarily small. For example, when the model is a linear model, the loss function is convex so any $\mu>0$ works for our algorithm. To verify this, in Figure 6 and Table 4 in Appendix E.6 in the rebuttal revision, we present the convergence curves and the test performance of our method when applied to training a linear model with $\mu$ of different values. The results show that $\mu$ can be as large as $\frac{10^{10}}{N+N-}\approx 1$ without impairing the convergence speed.
>
> **3. How this approach is meaningful from the theoretical viewpoint? Compared with the convergence analysis on deep neural networks plus $\frac{1}{\mu}\\|w\\|^2$ on the basis of convex analysis, how different this paper is?**
>
> **A:** We want to emphasize that our method and the convergence analysis are developed for the original non-convex DC function $F$. The quadratic term $\frac{1}{\mu}\\|v-w^k\\|^2$ we added is not just to make the subproblem convex but also to build a connection between the stationarity of the original function $F$ and the stationarity of the smooth approximation $F^{\mu}$. On the contrary, the method the reviewer mentioned (e.g. adding $\frac{1}{\mu}\\|w\\|^2$ to achieve weight-decay) targets on solving the convex regularized problem instead of the original non-convex problem.
>
>
> **4.Could the authors include the graph plots pAUC on a vertial axis and CPU+GPU time on a horizontal axis?**
>
> **A**: We include such graphs in Figure 7 in the Appendix. We also add $SVM\_{pAUC}^{tight}$'s pAUCs to the graph for comparison. It shows that our performance curves do not drop after exceeding the existing method's pAUCs. This should address the reviewer's concern.
>
> **5. Other than computation time, is it possible to report the number of epochs or iterations to exceed the $SVM\_{pAUC}^{tight}$'s pAUCs?**
>
> **A**: We add the number of epochs to exceed the $SVM\_{pAUC}^{tight}$'s pAUCs to *Table 1*. In fact, our method outperforms $SVM\_{pAUC}^{tight}$ in less than one epoch. This is because $SVM\_{pAUC}^{tight}$ does not produce a high-quality solution so it is not hard to outperform.

---

> > ### Comment · Reviewer_412K · 2022-08-09
> > **Thank your your answers.**
> >
> > Thank you for your answers. Some of my concerns are resolved. Therefore, I raised my score.
> > For Figure 7, could you please leave explanations about what GPU time is and how GPU was used in the implementation of $SVM_{pAUC}^{tight}$?

---

> > > ### Author Response · Authors · 2022-08-09
> > > **Response to Reviewer 412K**
> > >
> > > Dear Reviewer 412K,
> > >
> > > Thank you for reading our response and raising your score.
> > >
> > > The GPU time represents the time used if the algorithm is run on GPU. In Figure 7, the dashed line of $SVM\_{pAUC}^{tight}$ does not reflect its convergence with GPU time. It is reported for reference only since we use the authors’ MATLAB implementation which does not support GPU. We also upload revision with changed caption of Figure 7 to avoid confusion.
> > >
> > > Regards,
> > >
> > > Authors

---

> ### Author Response · Authors · 2022-08-08
> **To Reviewer 412K: Are your concerns addressed?**
>
> Dear Reviewer 412K,
>
> Please help check our responses to your questions/concerns and our updated paper, and let us know if you have any further questions.  We believe your mentioned weaknesses are not weakness of our paper but rather reflects the challenge of the problem (non-smooth non-convex DC programs), which we hope you can take into account for making the final decision. Thank you!
>
> Regards
> Authors

---

### Official Review · Reviewer_hV3Z · 2022-07-11

**Rating:** 6
**Confidence:** 4
**Soundness:** 2 fair
**Presentation:** 3 good
**Contribution:** 2 fair

**Summary:**

This paper proposes a gradient-based method for maximizing the partial AUC. The key steps are 1) decompose the objective as a difference-of-convex function, 2) smooth both convex components by their Moreau envelopes, and 3) perform (inexact) gradient-based updates on the smoothed DC objective function. In particular, gradients of the smoothed objective can be obtained through a lower-level stochastic block coordinate descent method.

**Questions:**

For numerical comparison, the proximal DCA seems to be a more relevant algorithm, which is essentially a proximal version of DCA. In analogues to subgradient method, DCA itself can be slow without further stabilization or acceleration. The authors are suggested to compare with related proximal DCA works.

**Limitations:**

The authors have admitted a limitation in the conclusion section. I do not see direct negative social impact from the current manuscript.

**Strengths And Weaknesses:**

1. Originality. The approach adopted in this paper seem to be new in the context of AUC maximization. However, components of the algorithmic design, i.e., smoothing of DC functions, and stochastic BCD, seem to be developed upon existing methodologies.
2. Quality. The overall quality is decent. The algorithmic design makes sense and the proof of Theorem 1 looks correct to me (admittedly I skipped the proofs of some intermediate results).
3. Clarity. The paper is well-organized and easy to follow.
4. Significance. The theoretical contribution is only mild. Numerical experiments seem promising.

---

> ### Author Response · Authors · 2022-08-02
> **Author response to Reviewer hV3Z**
>
> Thank you for your time and comments. The following are responses to your questions and concerns. The equation and reference numbers we mentioned below follow the rebuttal revision of the paper.
>
> **1. The theoretical contribution is only mild.**
>
> **A**: We are the first to show the theoretical complexity for finding a near critical point when both components in the DC program are **non-smooth**. Previous works either require one component to be smooth or assume a closed-form solution to (9). We agree that our work is built on several existing techniques, but there are some non-trivial proofs which were not seen in literature, for example, Lemma 5 and the proof of (30). We add Remark 1 in the revision to explain why (30) is important and non-trivial. Suppose we can set $T\_k$ in *lines 3 and 4 of Algorithm 2* appropriately such that the approximation errors $E\\| \bar{v}\_m^k - v\_{\mu f^m}(w^k)\\|^2$ and $E\\| \bar{v}\_n^k - v\_{\mu f^n}(w^k)\\|^2$ are both $O(1/k)$. We can then prove that *Algorithm 2* finds a nearly $\epsilon$-critical point within $K=\tilde O(\epsilon^{-2})$ iterations and the total complexity is $\sum\_{k=0}^{K-1}T\_k$. This is idea is not new. However, by *Proposition 1*, such a $T\_k$ must be $\Theta(k^2(\text{dist}^2(\bar\lambda^{(k)},\Lambda\_k^\*)+\\| v\_{\mu f^l}(w^{(k)})-w^{(k)}\\|^2))$ where $\text{dist}^2(\bar\lambda^{(k)},\Lambda\_k^\*)$ and $\\|v\_{\mu f^l}(w^{(k)})-w^{(k)}\\|^2$ also change with $k$. Then it is not clear what the order of $T\_k$ is. By a *novel* proving technique based on the (linear) error-bound condition of $g^l(w,\lambda)$ with respect to $\lambda$, we prove that both $\text{dist}^2(\bar\lambda^{(k)},\Lambda\_k^\*)$ and $\\|v\_{\mu f^l}(w^{(k)})-w^{(k)}\\|^2$ are $O(1)$ (see (27) and (30) in Appendix D) which ensures that $T\_k=\Theta(k^2)$ and thus the total complexity is $\sum_{k=0}^{K-1}T\_k=O(K^3)=\tilde O(\epsilon^{-6})$.
>
> **2. The algorithm is suggested to compare with related proximal DCA works.**
>
> **A**: We have added numerical comparison with proximal DCA by [62]. Please refer to Figure 1, Table 2 and Figure 3 in Appendix E.3 in rebuttal revision for details. From the graphs and table, we observe that our algorithm converges faster than proximal DCA and can achieve a better partial AUC. In fact, we observe that proximal DCA performs similarly to DCA on the testing instances. We also want to point out that the proximal DCA method by [62] will  converge in theory only when the first convex component is a smooth convex function plus a simple non-smooth convex function, which is not true in our problem. Hence, proximal DCA is not guaranteed to converge in theory for our instances.

---

### Official Review · Reviewer_AF28 · 2022-07-11

**Rating:** 7
**Confidence:** 4
**Soundness:** 3 good
**Presentation:** 3 good
**Contribution:** 3 good

**Summary:**

This work casts the partial AUC optimization problem into a non-smooth difference-of-convex (DC) program. Then, by means of the Moreau envelope smooth, this paper presents an effective approximated method that achieves the $\epsilon$-critical point with a $\tilde{\mathcal{O}}(1/\epsilon^6)$ complexity. Finally, empirical studies are conducted to show the strengths of the proposed approach.

**Questions:**

The primary concerns are two-fold:
- The strengths of the proposed algorithm on large-scale datasets are not clear. In general, solving the DC program is somewhat inefficient, and the authors should better provide some discussions about this.
- How to use the proposed DC-based optimization in the two-way partial AUC (TPAUC) problem? Because TPAUC is another practical generalization of AUC in real-world applications.

In addition, I hope the authors could also handle the weakness of this work listed in the above part.


**Strengths And Weaknesses:**

Strengths:
- New perspective and important problem. This paper addresses the partial AUC optimization from a novel point of view, i.e., a non-smooth DC problem.
- Roughly sound theoretical guarantee. The authors provide the convergence of the proposed algorithm, supporting the effectiveness theoretically.
- The writing and presentation are good.
Weakness:
- The related work seems not complete and some important literature in the AUC community is missing, such as
a)	Optimizing Two-way Partial AUC with an End-to-end Framework.
b)	Stochastic online AUC maximization.
c)	Learning with Multiclass AUC: Theory and Algorithms.
- The experiments seem not comprehensive. It is better to evaluate the performance on more long-tailed benchmark datasets, such as CIFAR-10-LT and TINY-IMAGENET-200-LT, and compare the proposed algorithms with SOTA partial AUC methods, i.e., [65] and [69] in their paper. Although extra experiments are not easy to complete, it is a relatively minor point considering the contribution of this work.

Overall, despite some issues, this work presents an interesting exploration of the partial AUC problem, which could motivate more effective methods and bring some potential value in this direction. Therefore, I tend to weak accept this paper and would improve my score if the authors could correct my concerns well.

---

> ### Author Response · Authors · 2022-08-02
> **Author response to Reviewer AF28**
>
> Thank you for your suggestions and comments. The following are our answers to your questions. The equation and reference numbers we mentioned below follow the rebuttal revision of the paper.
>
> **1. The related work seems not complete.**
>
> **A**: We have included the papers the reviewer mentioned in the section of related work in the rebuttal revision.
>
> **2. The experiments seem not comprehensive.**
>
> **A**: We do not have enough time to evaluate on the datasets suggested by the reviewer during the period of rebuttal. No matter the paper gets accepted, we will complete it in the next revision. Please notice that CheXpert is a highly imbalanced medical dataset with *positive percentage = 0.12*. Concerning the comparison with SOTA partial AUC methods, we are not able to compare with the algorithms mentioned by the reviewer because those algorithms can only be applied when the partial AUC is defined with FPR in a range $[0,\beta]$, but in our problem, FPR is in a range $[\alpha,\beta]$ with $\alpha > 0$ which makes the optimization model fundamentally different.
>
>
> **3. The strengths of the proposed algorithm on large-scale datasets are not clear.**
>
> **A**: A DC program itself does not always means inefficiency. It depends on the algorithm people use. Directly applying a DC algorithm such as [24] to DC programs (5) or (10) is indeed not efficient for large-scale datasets because it needs to process the entire dataset in each main iteration to compute the subgradients of $f^m$. However, based on reformulation (12), our method only needs to perform stochastic coordinate subgradient updates that only requires a small sample of data in each iteration. This guarantees the scalability of our method on large datasets (e.g., the CheXpert dataset used in the paper has over 200K images).
>
> **4. How to use the proposed DC-based optimization in the two-way partial AUC (TPAUC) problem?**
>
> **A**: Our DC-based optimization method cannot be directly applied to the TPAUC problem although it is not difficult to modify it for TPAUC, for example, by setting $m=0$ and introducing a dual variable to represent the top-$k$ losses over positive data (similar to Appendix C in [72]).  A few numerical methods have been proposed for TPAUC problem such as [64,66,67,72]. Although we study one-way partial AUC, those methods cannot be applied to our problem at all because their partial AUC is defined with FPR in a range $[0,\beta]$, but in our problem, FPR is in a range $[\alpha,\beta]$ with $\alpha > 0$. This makes our problem a non-smooth DC problem, which is fundamentally more difficult than the TPAUC optimization considered in those works.

---

> > ### Comment · Reviewer_AF28 · 2022-08-08
> > **Response to the authors' feedback**
> >
> > Thank you so much for your replies! I think the author gives good explanations for my concerns. I understand that some of the experiments cannot be done right now due to the time limit. Balancing Pros and cons, I think this is an interesting paper with a novel technique to solve the difficult problem of Partial AUC optimization. Those said, I would raise my score to support accepting this paper. Nonetheless, I hope the authors can attach the new experimental results in the final version as they promised.

---

> > > ### Author Response · Authors · 2022-08-08
> > > **Thank you!**
> > >
> > > Dear Reviewer AF28,
> > >
> > > Thank you for agreeing that our paper gives a novel technique for partial AUC optimization. We will add new experimental results in the final version.
> > >
> > > Regards
> > > Authors

---

### Official Review · Reviewer_6ygH · 2022-07-13

**Rating:** 5
**Confidence:** 3
**Soundness:** 2 fair
**Presentation:** 3 good
**Contribution:** 2 fair

**Summary:**

This paper aims to solve the scalability issue of partial AUC maximization (including the sum of ranked range loss minimization) in the setting of large-scale imbalanced binary classification. It proposes a gradient-based optimization algorithm with (non-asymptotic) convergence analyses. Specifically, it casts the (surrogate) partial AUC maximization problem into a non-smooth difference-of-convex (DC) program and then develops  an approximated gradient descent method based on the Moreau envelope smoothing technique. Besides, it also uses a stochastic block coordinate update to get the stochastic gradient in the algorithm. Finally, experimental results are provided.

**Questions:**

1. As mentioned by the authors, the convergence analysis of the proposed method is the main contribution. Then, what are the technical novelty and its challenge?

**Limitations:**

Although the authors have mentioned the limitations of this work, I have the following suggestion.
1. As mentioned by the authors, the convergence analysis of the proposed method is the main contribution. Thus it is better to summarize the technical novelty and its challenge. Besides, synthetic experimental results can be added to illustrate the effectiveness of the convergence rate obtained by the theoretical results in this paper. Moreover, it is convincing to compare with existing theory results.
2. There are many typos in this manuscript. For example,
  -line 230, According to Theorem 1.
  -line 292, Figure2...
Please check and fix them carefully.


**Strengths And Weaknesses:**

First of all, I have to admit that I am not an expert in this area and have not checked the proofs in detail. Below are my comments.

Strengths:

1. Clarity.

Overall, this paper is written clearly.

2.  Quality.

It is good that the theoretical analysis (i.e. the convergence rate analysis) and experimental results are both provided to illustrate the effectiveness of the proposed optimization algorithm for partial AUC maximization.

Weaknesses:

1. Originality.

I am afraid that the novelty is limited since the techniques such as the difference-of-convex (DC) program transformation and related smoothing techniques have been widely studied in the related literature although there is little difference in the setting (i.e., both components in the DC program are non-smooth) as mentioned by the authors. Besides, it is incremental to the DCA method.

2. Significance.

For the proposed optimization algorithm, there are many hyperparameters, which are hard to tune in practice. Thus, I am concerned that it is not practical in applications. For the proof technique of theory, I cannot see the novelty. Thus, I am afraid that this work may have little influence on the community.

3. Quality.

The proposed method is based on the widely-used techniques in the community. Besides, the claimed main contribution (i.e. theoretical results) does not compare with the existing theoretical results.

---

> ### Author Response · Authors · 2022-08-02
> **Author response to Reviewer 6ygH**
>
> Thank you for your time and comments. The following are responses to your questions and concerns. The equation and reference numbers we mentioned follow the rebuttal revision of the paper.
>
> **1. The novelty is limited and it is incremental to the DCA method.**
>
> **A**: We politely disagree with this statement. With both $f^n$ and $f^m$ being non-smooth, the difficulty of the DC problem increases to a new level. In this case, before our work, it was not clear what notion of stationary point an algorithm can converge to in a non-asymptotic way. In fact, all papers before our work such as [52] must require at least one of $f^n$ and $f^m$ to be smooth in order to establish some convergence guarantee. Proposing a new definition of a near critical point, our work is the first one that can theoretically guarantee convergence in the fully non-smooth case. Moreover, the DCA method, e.g., [24], is not efficient for large data because it needs to compute the deterministic subgradient of $f^m$, which requires processing the entire dataset. On the contrary, we propose a reformulation that allows us to construct a stochastic subgradient of $g^l$ and apply coordinate update to $\lambda$ by sampling data. This makes our method much more efficient than the DCA method for pAUC maximization on large data.
>
> **2. There are so many hyperparameters that are hard to tune in practice.**
>
> **A**: There are four hyperparameters, $c$, $T\_k$, $\gamma$ and $\mu$, to tune, but it is not hard to find the good values or ranges for them, at least for the datasets we test on. For $\gamma$ and $\mu$, we found through experiments that choosing a value from just two candidates is enough to guarantee a good performance. We also found that $T_k=50(k+1)^2$ works well for multiple datasets. Parameter $c$ is the only one that needs to be tuned over a relatively large grid. We suggest users in practices to follow our tuning process, which should work for most datasets with a reasonable effort of tuning.
>
> **3. Theoretical results are not compared with the existing theoretical results.**
>
> **A**: We consider a DC problem with both components non-smooth. To the best of our knowledge, there are no existing algorithms that can solve this problem with theoretically guaranteed complexity. Therefore, there are no existing theoretical results we can compare with.
>
> **4. What are the technical novelty and its challenge?**
>
> **A**: Although Moreau envelope and stochastic coordinate descent are well-known techniques, there are some non-trivial technical proofs which were not seen in literature before, for example, Lemma 5 and the proof of (30). We add Remark 1 in the revision to explain why (30) is important and non-trivial. Suppose we can set $T\_k$ in *lines 3 and 4 of Algorithm 2* appropriately such that the approximation errors $E\\| \bar{v}\_m^k - v\_{\mu f^m}(w^k)\\|^2$ and $E\\| \bar{v}\_n^k - v\_{\mu f^n}(w^k)\\|^2$ are both $O(1/k)$. We can then prove that *Algorithm 2* finds a nearly $\epsilon$-critical point within $K=\tilde O(\epsilon^{-2})$ iterations and the total complexity is $\sum\_{k=0}^{K-1}T\_k$. This idea is not new. However, by *Proposition 1*, such a $T\_k$ must be $\Theta(k^2(\text{dist}^2(\bar\lambda^{(k)},\Lambda\_k^\*)+\\| v\_{\mu f^l}(w^{(k)})-w^{(k)}\\|^2))$ where $\text{dist}^2(\bar\lambda^{(k)},\Lambda\_k^\*)$ and $\\|v\_{\mu f^l}(w^{(k)})-w^{(k)}\\|^2$ also change with $k$. Then it is not clear what the order of $T\_k$ is. By a *novel* proving technique based on the (linear) error-bound condition of $g^l(w,\lambda)$ with respect to $\lambda$, we prove that both $\text{dist}^2(\bar\lambda^{(k)},\Lambda\_k^\*)$ and $\\|v\_{\mu f^l}(w^{(k)})-w^{(k)}\\|^2$ are $O(1)$ (see (27) and (30) in Appendix D) which ensures that $T\_k=\Theta(k^2)$ and thus the total complexity is $\sum_{k=0}^{K-1}T\_k=O(K^3)=\tilde O(\epsilon^{-6})$.

---

> ### Author Response · Authors · 2022-08-08
> **To Reviewer 6ygH**
>
> Dear Reviewer 6ygH,
>
> Please help check our responses and let us know if you have any further questions. We believe that after reading our responses, you will better understand the challenges and significance of our results. Thank you!
>
> Regards
> Authors

---

### Meta-Review · Area_Chair_f3J9 · 2022-08-26

**Recommendation:** Accept
**Confidence:** Certain

**Metareview:**

The paper proposes a difference-of-convex (DC) based algorithm for optimizing the partial AUC performance measure. The authors show convergence guarantees and superior empirical performance compared to baselines.

The reviewers had some concern about the novelty of the use of DC optimization, to which the authors point out that the presence of non-smooth terms in both components of the DC objective makes their formulation non-standard and interesting.

Overall, the reviewers seem to agree that paper should be accepted.

To address some pending concerns, the authors are **strongly encouraged** to include the following:
- A detailed discussion on hyper-parameter tuning and the values they prescribe in practice, as well as, an analysis of how sensitive their algorithm is to the hyper-parameters
- The additional experiments that the authors promised to Reviewer AF28

We trust that the authors will include the above in the camera-ready version of the paper.

**Award:**

No

---

### Decision · Program_Chairs · 2022-09-14

Accept